# A Novel Method for Rapid and High-Performance SERS Substrate Fabrication by Combination of Cold Plasma and Laser Treatment

**DOI:** 10.3390/nano14211689

**Published:** 2024-10-22

**Authors:** Thi Quynh Xuan Le, Thanh Binh Pham, Van Chuc Nguyen, Minh Thu Nguyen, Thu Loan Nguyen, Nguyen Thuan Dao

**Affiliations:** 1Institute of Materials Science (IMS), Vietnam Academy of Science and Technology (VAST), 18 Hoang Quoc Viet, Cau Giay, Hanoi 100000, Vietnam; xuanltq@ims.vast.ac.vn (T.Q.X.L.); binhpt@ims.vast.ac.vn (T.B.P.); chucnv@ims.vast.ac.vn (V.C.N.); thunm@ims.vast.ac.vn (M.T.N.); loannt@ims.vast.ac.vn (T.L.N.); 2Graduate University of Science and Technology (GUST), Vietnam Academy of Science and Technology (VAST), 18 Hoang Quoc Viet, Cau Giay, Hanoi 100000, Vietnam

**Keywords:** nanomaterials, gold nanoparticles, SERS, cold plasma, laser

## Abstract

In this paper, we report a simple yet efficient method for rapid and high-performance SERS substrate fabrication by a combination of cold plasma and laser treatment. Our analysis reveals that cold plasma pre-treatment significantly reduced surface roughness, transforming 200 nm spikes into an almost perfectly uniform surface, while enhancing the substrate’s surface energy by lowering the water contact angle from 59° to 0°, all achieved within just 30 s of 0.9-mW plasma treatment, while 15-min green-laser treatment facilitated more uniform deposition of AuNPs across the entire treated area, effectively creating the SERS substrates. The combined treatments result in enhancement of the Raman intensity (11 times) and consistency over the whole area of the SERS substrates, and their reusability (up to 10 times). The fabricated SERS substrates exhibit a significant enhancement factor of approximately 3 × 10⁸ with R6G, allowing detection down to a concentration of 10^−12^ M. We demonstrate the application of these SERS substrates by detecting amoxicillin—an antibiotic used worldwide to treat a diversity of bacterial infections—in a dynamic expanded linear range of seven orders (from 10^−3^ to 10^−9^ M) with high reliability (R^2^ = 0.98), and a detection limit of 9 × 10^−10^ M. Our approach to high-performance SERS substrate fabrication holds potential for further expansion to other metallic NPs like Ag, or magnetic NPs (Fe_3_O_4_).

## 1. Introduction

Raman scattering, discovered by C.V. Raman in 1928 [1], evolved into a spectroscopic technique used to investigate the chemical composition and structure of molecules and crystals [2]. In Raman spectroscopy, a monochromatic laser excites molecules in a sample, causing them to vibrate and emit scattered photons. A small fraction of these photons has different energy levels, known as Raman scattering, which reveals the vibrational energy of the molecules and provides insights into their bonding, symmetry, and crystal structure. Surface-enhanced Raman scattering (SERS) is an advanced Raman technique known for its high sensitivity in the analysis of chemicals and molecules [3,4,5]. The SERS enhancement mechanism primarily involves (1) electromagnetic field (EMF) enhancement, where surface plasmon resonance on a metal surface amplifies the Raman scattering signal, and (2) chemical enhancement, where metal particles and molecules exchanges their polarization and vibrational frequency by interactions like charge transfer or chemical bonding [6]. Together, these mechanisms make SERS an exceptionally sensitive technique capable of detecting extremely low quantity of molecules, down to a single molecule [7], across various scientific fields, including analytical chemistry [8], food safety [9], environmental monitoring [10], and biochemistry [11]. Surface plasmon resonance is frequently used in SERS, which increases the spontaneous Raman scattering of appropriate molecules by creating an increased field in the metal structure [12,13,14].

Noble metal nanoparticles, such as gold (AuNPs) and silver (AgNPs), enhance Raman scattering through their plasmonic properties, and AuNPs have been researched extensively for their compatibility with biomolecules and potential applications in diagnostics and therapeutics [15]. AuNPs are most effective when around 45–50 nm in diameter due to optimal surface concentration and area, and they exhibit minimal toxicity when combined with biomolecules at biological concentrations [16]. In our previous work, we utilized a cold atmospheric pressure plasma jet technique to synthesize stabilizer-free, homogeneous AuNPs with an average diameter of approximately 45 nm, demonstrating remarkable optical sensing properties compared to commercial AuNPs, and offering potential for the development of versatile and highly sensitive biosensors [17].

Cold atmospheric plasma (CAP) is a partially ionized gas produced at near-room temperature under atmospheric pressure, containing a mix of electrons, UV radiation, ions, reactive species, and neutral particles [18]. In surface treatment, CAP is employed to enhance surface properties such as hydrophilicity, adhesion, and sterilization by altering surface chemistry, reducing surface roughness, and removing organic contaminants [19]. There have been a few reports of attempts to apply CAP in SERS for poly methyl methacrylate (PMMA) to clean and activate the substrate [20], or regenerate reusable SERS substrates [21]. Although, in practical terms, approaches such as these can contribute to a reduction of the detection costs, no significant enhancement in performance and sensitivity of the SERS substate can be achieved with these treatments.

Laser deposition of noble metal (Au/Ag) nanoparticles has been utilized to create SERS substrates by directing a laser onto a metal-containing target, causing localized heating and ablation, resulting in the formation of Au/Ag NPs that enhance Raman signals [22,23,24,25]. The size, shape, and distribution of the NPs can be finely tuned by adjusting the laser parameters, enabling precise control over the SERS substrate’s enhancement capabilities. In most of the cases, the metal nanoparticles were formed in-situ under the influence of laser ablation; hence, this method has often required expensive pulsed [22,24] or even ultrashort-pulse [23] lasers to precisely control the formation of the nanoparticles. Recently, a study reported the use of an argon cold plasma jet combined with nanosecond laser ablation (248 nm, 25 ns, 10 Hz) to deposit silver aerosols onto a substrate at room pressure [26]. However, the deposited particles were clustered; hence, the SERS signals were not reproducible due to the large spatial variation of the particle nanofeatures as well as surface coverage. Our recent work—in planting Ag nano-dendrites at the spheroid end-facet of an optical fiber, assisted by a dual-laser to make a SERS substrate—resulted in an enhancement factor EF of ~2.54 × 10^7^, which surpasses that of most other Au/Ag NP-based SERS substrates [27]. While making a SERS substrate on the tip of an optical fiber provides a versatile and rapid detection method, its practicality is limited by the difficulty of cleaning and the high cost of the optical fiber.

Numerous studies have explored Au nanoparticle-based SERS substrates using a variety of preparation methods. These range from the simplest, such as dropping an AuNP mixture onto a silicon substrate [28], to more complex approaches like reducing HAuCl_4_ on a polymer substrate [29], or using electrochemically roughened nano-Au films [30]. Advanced methods, such as patterned assemblies of Au NPs based on polymer brush templates [31], have also been developed. The enhancement factor, depending on the probe molecules used, varies significantly from 10^5^ to 10^9^. In general, more advanced and complex fabrication techniques, which involve more expensive equipment—offering greater precision and resulting in a closely packed, uniform assembly of AuNPs on the substrate—lead to higher enhancement factors.

In this study, we attempted a combination of cold plasma and laser treatment for the development of a simple, low-cost yet efficient method for rapid and high-performance SERS substrate fabrication. We performed a detailed analysis by AFM, SEM, and water contact angle to realize the effect each treatment contributed to the enhancement of the substrate’s surface. And we validated various high-performance properties: the enhancement of the Raman intensity, the consistency over a large area, high reproducibility, and the reusability of the fabricated SERS substrate. Finally, we demonstrated the extended linear range and high sensitivity of the fabricated SERS substrate using R6G and highlighted its potential in detecting amoxicillin, an antibiotic commonly used to treat a variety of bacterial infections. Our result offers a quick, low cost, and efficient approach for fabricating a high-performance AuNP-based SERS substrate, with the potential for further extension to other metallic NPs like Ag, or magnetic NPs (Fe_3_O_4_).

## 2. Materials and Methods

### 2.1. Chemicals

Gold nanoparticles (AuNPs) were synthesized by the plasma jet method from gold (III) chloride trihydrate (HAuCl_4_.3H_2_O, Sigma Aldrich, Ltd., Singapore) in double-distilled water as reported in [17]. We would like to emphasize that the synthesis processes was stabilizer-free; hence, there were no capping agents on the surface of these nanoparticles. The synthesized AuNPs with an average diameter of ~45 nm and 2.6 OD (optical density) were stored in stock at 22–25 °C. Rhodamine 6G (R6G, Sigma Aldrich, Ltd., Singapore) and Amoxicillin (C_16_H_19_N_3_O_5_S) were purchased from Sigma Aldrich, Ltd., Singapore and used without further purification.

### 2.2. Fabrication Processes of the SERS Substrate

To fabricate the SERS substrate used in this study, we used a combination process of cold plasma and laser treatment as illustrated in Figure 1. Glass slides (thickness 1.0 mm, 75 × 25 mm, Avantor, Ltd., Singapore) were cut to 8 × 8 mm size. They were cleaned in several steps with distilled water, ethanol, and acetone, then distilled water again; dried by nitrogen gas; and then stored at room temperature. A 6 mm diameter hole in polymer cover tape was attached to a clean glass slide to create an area ~30 mm^2^. It was pre-treated using a cold plasma jet (0.9 W) for 30 s. The setup of the plasma jet was described in our previous publication [32]. Briefly, a tunable high voltage–frequency power supply (2–6 kV and 100 Hz–80 kHz) was used to generate plasma in a quartz capillary tube, with a stainless-steel syringe as an inner electrode and a copper electrode wrapped around the outside. The plasma jet emitted from this setup had a maximum length of 1.5–1.8 cm, with a distance of ~0.5 cm from the nozzle to the substrate’s surface. A volume of 3 µL gold nanoparticles with a concentration of 1.2 OD was dropped on the confined area. The area was then exposed to a diode green 532 nm laser (Laserlands, Wuhan, China) with an optical power of laser beam illumination of 480 mW for 15 min to assist the deposition of AuNPs to fabricate the SERS substrate. The laser beam was focused onto one end of a multimode optical fiber (105/125 µm) (Thorlabs, NZ, USA) using a 10× objective lens (NA = 0.22) (Thorlabs, NZ, USA), which was attached to the GRIN-rod (Lightglass Optics, NM, USA).

### 2.3. Characterization Techniques for the AuNPs, Cold Plasma, Laser, and SERS Substrate

*UV absorption of the AuNPs:* UV absorption measurements were made using a custom-built Ocean Optics UV/VIS system using quartz cuvettes with a 10 mm optical path length and an internal width of 2 mm, requiring a minimum sample volume of 300 µL. The UV/VIS system consisted of a DH-2000-BAL light source, USB4000 spectrometer (Ocean Optics, Inc., Orlando, FL, USA) and a cuvette holder, all connected via optical fibers. Data acquisition was performed using Oceanview software (version 2.0.8), with three scans collected and averaged for each measurement.

*Electrical properties of the cold plasma:* The power source’s output voltage and current waveforms of the plasma system were measured using a three-component Tektronix system: a TBS1154 oscilloscope paired with a P6015 voltage probe (1000:1) and a P6021A current probe.

*Optical Emission Spectrum (OES) of the cold plasma jet:* The optical emission from the plasma jet discharge was collected 5 mm from the nozzle by an optical fiber connected to the input of a high-resolution Aurora4000 spectrometer (PhotonTec Berlin GmbH, Germany). Spectra were scanned with a 2 s integration time, using an entrance aperture of 0.05 mm. In all cases, the emission spectrum was obtained by acquiring and averaging three continuous scans.

*Atomic Force Microscope (AFM):* The thickness of the AuNP layer on the SERS substrate was estimated by AFM. AFM images were recorded on a PicoScan 2500 (Agilent Technologies, Santa Clara, CA, USA) in tapping mode using a long cantilever Tap 190Al-G probe (Budget Sensors, Bulgaria) (resonance frequency f = 190 kHz, force constant k = 48 N·m^−1^).

*Scanning Electron Microscopy (SEM):* A Hitachi S-4800, Japan field emission scanning electron microscope operated at 5 kV acceleration voltage was used to explore the surface morphology and uniformity of the SERS substrates.

*Raman Spectroscopy*: Different concentrations of R6G or Amoxicillin were prepared in DI water, and 10 µL of each solution was applied onto SERS substrates. The SERS spectra were recorded using a HORIBA XploRA™ PLUS Raman spectrometer (HORIBA France SAS, Longjumeau, France) with a 532 nm laser, 30 s acquisition time, 1200-grating, and 100 µm pinhole diameter. The experiment was conducted with an optical density filter of 0.1 or 0.01, and the Raman spectra at each concentration were averaged from measurements taken at five different spots, unless otherwise stated.

## 3. Results and Discussion

### 3.1. Effect of Cold Plasma Pre-Treatment on the Glass Surface

Recently, non-thermal plasma has been employed as a quick and efficient method for cleaning and modifying the surface of materials [33,34]. Here, we applied cold plasma treatment to the surface of the glass substrate by projecting a plasma beam on the whole surface limited by a polymer cover tape. Figure 2A shows the current–voltage profile of the plasma discharge with a peak-to-peak voltage of ~3.5 kV, a peak current of ~20 mA, and a frequency of ~45 kHz. The estimated power of the plasma jet was 0.9 W. To examine the reactive plasma species, we measured the plasma emission spectrum in a full-range 200–1000 nm region (Figure 2B). The 309 nm emission peak, indicative of hydroxyl radicals (OH), resulted from the dissociation of water vapor in ambient air, while the weak emission lines detected in the 330–380 nm region belonged to nitrogen [17]. Emission peaks in the 650–1000 nm range were attributed to transitions from excited argon (Ar) levels to metastable states, because we used argon as the flowing gas. Reactions with plasma species, particularly hydroxyl radicals, on the glass surface, lead to the formation of various chemical groups, while electron and ion bombardment from the plasma can alter the surface morphology [35].

Figure 2C–E present the effect of different plasma treatment time on the glass surface: before (C), after 10 s (D), and 30 s (E) of plasma treatment. It distinctly shows two effects of plasma treatment: (1) reducing the roughness of the glass surface (from 200 nm before plasma treatment to an almost ideally uniform surface after 30 s of plasma treatment); and (2) enhancing surface energy by reducing the water contact angle from 59° to 26° and close to 0° with 0, 10, and 30 s of plasma treatment time, respectively. These two effects improved the surface quality for better deposition of the AuNPs on this surface in the AuNP deposition step to make the SERS substrate.

### 3.2. Effect of Laser Treatment on the Distribution of the AuNPs on the Surface

After plasma pre-treatment, a volume of AuNPs was dropped on the confined area, followed by 15-min green laser treatment to create the SERS substrate. We used AuNPs, synthesized by cold plasma jets as described in [17], with an average diameter of 45 nm, an absorption peak at 535 nm, and a narrow full-width-half-maximum (FWHM) of only 84 nm (Figure 3A). We also conducted UV-VIS spectral analysis on the relevant substrates: bare glass, plasma-treated glass, and AuNPs deposited via laser on plasma-treated glass, as shown in Appendix A. The absorption band around 530 nm for the fabricated SERS substrate aligns with the absorption peaks of the AuNPs and the 532 nm excitation laser used in Raman measurements. This alignment facilitates the surface plasmon effect, allowing high SERS performance. Meanwhile, the AFM image reveals that, under the assistance of the green laser treatment, AuNPs were deposited uniformly on the glass surface, forming a few layers with a total thickness less than 150 nm (×3 times the diameter of an average particle) as shown in Figure 3B. To determine the effectiveness of the laser treatment, the SEM image of AuNPs on the glass surface with laser treatment was compared with the one without laser treatment. AuNPs were deposited uniformly on the glass surface with the assistance of the green laser treatment (Figure 3C), while there was mixing of some islands of the AuNPs with hollow areas on the surface of the glass substrate without laser treatment (Figure 3D). In our fabrication approach, key factors such as inter-particle spacing, the number of particle layers, and the degree of aggregation determine the quality of the SERS effect. We experimented with different concentrations, volumes of AuNPs, and laser exposure times to identify the optimal parameters: 3 µL of AuNPs with a concentration of 1.2 OD and a 15-min laser exposure. These settings resulted in a uniform deposition and a closely packed arrangement of AuNPs on the substrate, as shown in the SEM image (Figure 3C), with fewer than four layers (under 200 nm), as confirmed by the 3D AFM image (Figure 3B).

### 3.3. Effect of the Combination of Cold Plasma and Laser Treatment

#### 3.3.1. Enhancement of the Raman Intensity and Consistency over a Large Area

To evaluate the efficacy of the cold plasma pre-treatment and laser treatment on the SERS substrate, we compared the Raman spectra of R6G on different SERS substrates: those fabricated without any treatment (black), with plasma treatment but no laser (violet), and with both plasma and laser treatments (green), as shown in Figure 4A. These spectra were measured at 11 different spots distributed across the entire substrate’s surface (Figure 4B). For better illustration, the peaks at 613 cm^−1^ and 1361 cm^−1^ were highlighted by dots, while the error bars represented the deviation of 11 measurements. By comparing a signature Raman peak of R6G at 1361 cm^−1^ [36], we observed that Raman intensity can be increased 3× times by plasma pre-treatment, from I_(no plasma + no laser)_~1000 a.u. to I_(plasma + no laser)_~3000 a.u., and up to 11× times by combination of both plasma and laser treatment, I_(plasma + laser)_~11,000 a.u. We also want to emphasize that the Raman signature peaks of R6G (e.g., at 613 cm^−1^ and 1361 cm^−1^) were unchanged for all three SERS substrates, indicating that both cold plasma pre-treatment and laser treatment did not affect the position of the Raman peaks of the study materials. Moreover, not only the Raman intensity but also the uniformity of the SERS substrate was improved as displayed by the shorter error bars between these samples, respectively. We have demonstrated that the combination of plasma and laser treatment has improved the surface’s quality and produced a more uniform distribution of AuNPs on the substrate (Figure 3), leading to enhanced Raman intensity and a more uniform SERS substrate and, hence, more consistent SERS signals (Figure 4).

#### 3.3.2. Improved Reusability of the SERS Substrate

The reusability of the SERS substrate was tested using R6G solutions at a concentration of 10^−6^ M. After recording the Raman spectrum, the substrates were cleaned by immersing them in distilled water for 1 min, followed by ethanol for 10 s, acetone for 10 s, and then distilled water again for 10 min to remove the remaining molecules, before being dried with nitrogen gas. We recorded the Raman spectrum of the cleaned substrate before re-adsorbing the analyte, and this process was repeated for 10 cycles of washing and re-adsorption. No R6G was detected in the Raman spectra after washing, indicating complete removal of the remaining analyte molecules. Figure 5 shows the Raman spectra of R6G for freshly made substrates and after the first, third, fifth, and tenth re-adsorptions, comparing (plasma + laser)-treated SERS substrates (Figure 5A) with those pre-treated only with cold plasma (Figure 5B). While the Raman intensity gradually decreased with each washing cycle, the (plasma + laser)-treated substrate still provided a distinct spectrum after 10 cycles, whereas the (only cold plasma)-treated substrate showed almost no signal after 5 cycles. Moreover, we observed that the Raman intensity of R6G on the fabricated SERS substrate remained at 85% after 3 months stored at room temperature (Appendix A). This demonstrates the stability, reusability, and cost-effectiveness of our (plasma + laser)-treated SERS substrate.

#### 3.3.3. SERS Enhancement Factor (EF) and Sensitivity with R6G

To investigate the Raman signal enhancement, R6G also was employed as a standard Raman reporter [37]. For simplicity, an enhancement factor (*EF*) was calculated as follows:(1)$$EF=\frac{I_{SERS}}{I_{R}}/\frac{C_{SERS}}{C_{R}}$$
where *c_R_* and *c_SERS_* are R6G concentrations at a bare substrate (normal Raman) and SERS measurements, and *I_R_* and *I_SERS_* are their peak intensities, respectively. Figure 6 shows the SERS spectrum of R6G at an extreme low concentration of 10^−12^ M, measured with our fabricated SERS substrate. Signature peaks of R6G can be observed: C–C–C ring in-plane bending at 613 cm^−1^, C–H out-of-plane bending at 775 cm^−1^, C–H in-plane bending at 1185 cm^−1^, and aromatic C–C stretching at 1312, 1361, 1509, 1574, and 1650 cm^−1^ [36].

Using Raman signals of R6G at 10^−12^ M on the SERS substrate and 10^−3^ M on the glass substrate, we calculated the enhancement factor:(2)$$EF_{R6G}=\frac{I_{SERS}}{I_{R}}/\frac{C_{SERS}}{C_{R}}=\frac{217}{667}\frac{10^{-3}}{10^{-12}}=3.2\times 10^{8}$$

Thanks to the uniformly distributed AuNPs on the surface from plasma and laser treatments, our SERS substrate has achieved a significant *EF* of 3.2 × 10^8^, much better than other work on SERS substrates with dispersion of AuNPs (*EF*~10^5^) [28], or reducing Au ions with silicon nanocrystal-containing polymer microspheres (*EF*~5.4 × 10^7^) [29]. Our SERS’s *EF* is similar to the ones produced by more complicated and multiple-process methods, such as electrochemically roughened nano-Au film (*EF*~2.45 × 10^8^) [38] or wool roll-like Ag nanoflowers by mixed ethanol–water reaction (*EF*~2.7 × 10^6^–5.4 × 10^9^) [30], as shown in Table 1.

### 3.4. Application of the Fabricated SERS Substrate to Detect Amoxicillin

We applied the fabricated SERS substrate to detect amoxicillin, a widely used antibiotic in both humans and food-producing animals. Due to concerns about potential cancer-causing effects and the ability of high-dose amoxicillin residues in animal products to cause gene mutations, there is a pressing need for a quick, simple, and accurate detection method. Figure 7A shows SERS spectra of amoxicillin measured at different concentrations from 10^−3^ to 10^−9^ M, showing signature peaks at 1456 cm^−1^, 1276 cm^−1^, and 852 cm^−1^ [40,41]. Using Raman signals of amoxicillin at 10^−6^ M on the SERS substrate and 1 M on the glass substrate, we calculated the enhancement factor:(3)$$EF_{\mathrm{amoxicillin}}=\frac{I_{SERS}}{I_{R}}/\frac{C_{SERS}}{C_{R}}=\frac{161}{1887}\frac{1}{10^{-9}}=8.5\times 10^{7}$$

The linear correlation between Raman intensity and amoxicillin concentration spans across seven orders of magnitude (from 10^−3^ to 10^−9^ M), showing a very high reliability with an *R*^2^ value of 0.98 (Figure 7B). The limit of detection (LOD) was determined to be 0.9 × 10^−9^, calculated based on the linear correlation at an intensity three times that of the blank noise [38].

## 4. Conclusions

In this work, we present a simple yet efficient method for fabricating high-performance SERS substrates through a combination of cold plasma and laser treatment. Our analysis reveals that cold plasma pre-treatment reduces surface roughness and increases surface energy, while laser treatment ensures a more uniform deposition of AuNPs, resulting in enhanced SERS substrates. This combined approach leads to an 11-fold increase in Raman intensity, improved consistency across the whole substrate, and reusability for up to 10 cycles. The fabricated SERS substrates exhibit an excellent enhancement factor (~3 × 10^8^ with R6G) and a remarkable detection limit of 10^−12^ M. We applied these substrates to detect amoxicillin, a widely used antibiotic, achieving a broad linear detection range across seven orders of magnitude (from 10^−3^ to 10^−9^ M) with excellent reliability (R^2^ = 0.98) and a detection limit of 9 × 10^−10^ M. Our high-performance SERS substrate fabrication approach has the potential for further expansion using more cost-effective metallic nanoparticles like Ag or magnetic nanoparticles such as Fe_3_O_4_. This combined approach provides enhanced sensitivity, uniformity in detecting trace molecules, and cost-effective reusability and, hence, holds great promise for versatility in chemical sensing, environmental monitoring, and biomedical diagnostics.

## Figures and Tables

**Figure 1 nanomaterials-14-01689-f001:**
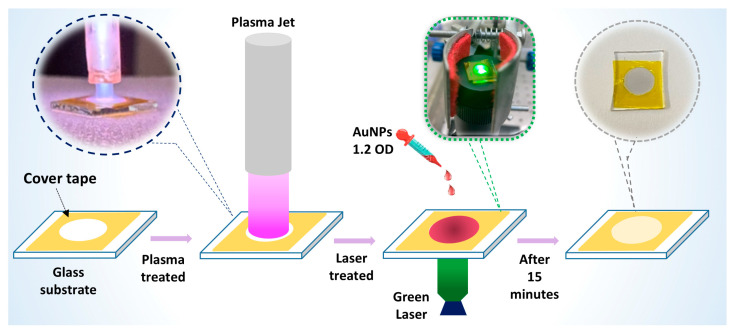
SERS substrate fabrication process: A 6 mm diameter hole in polymer cover tape was attached to a clean glass slide to create an area ~30 mm^2^. It was pre-treated by a cold plasma jet for 30 s. A volume of gold nanoparticles (diameter ~50 nm) was dropped on the confined area, followed by 15-min green laser treatment to create the SERS substrate.

**Figure 2 nanomaterials-14-01689-f002:**
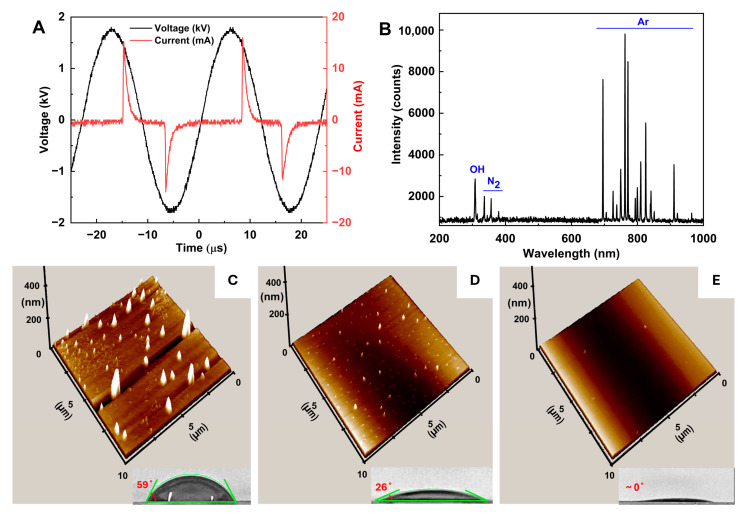
Characteristics of the plasma jet: (**A**) voltage–current waveform and (**B**) emission spectrum. (**C**–**E**) AFM images of the glass surface and photos of the water contact angle (bottom right) before (**C**), after 10 s (**D**), and 30 s (**E**) of plasma treatment.

**Figure 3 nanomaterials-14-01689-f003:**
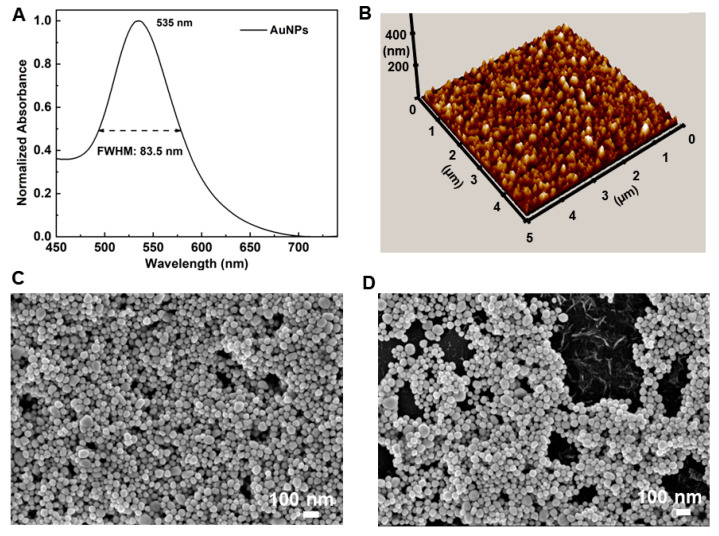
(**A**) Absorption spectrum of the AuNPs. (**B**) AFM profile of the AuNPs on the glass surface. (**C**,**D**) SEM images of the AuNPs on the glass surface with laser treatment (**C**) and without laser treatment (**D**).

**Figure 4 nanomaterials-14-01689-f004:**
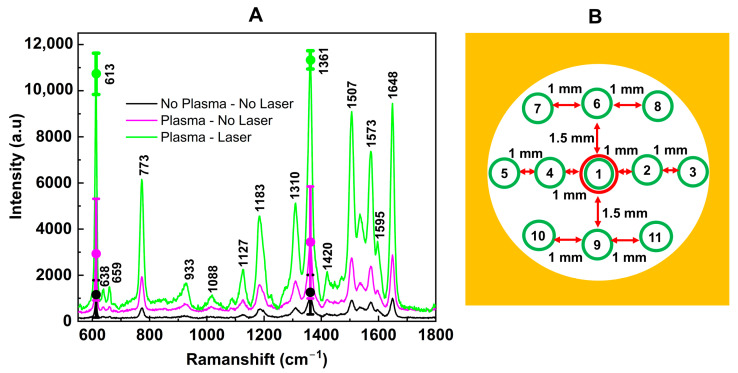
(**A**) Raman spectra of R6G measured on SERS substrate fabricated without either cold plasma or laser treatment (black), with only cold plasma pre-treatment (pink), and with both cold plasma and laser treatment (green). Each Raman spectrum is an average of 11 measurements at 11 different spots distributed on the entire substrate’s surface (**B**), and their deviation is displayed by the error bar on the signature peaks at 613 cm^−1^ and 1361 cm^−1^.

**Figure 5 nanomaterials-14-01689-f005:**
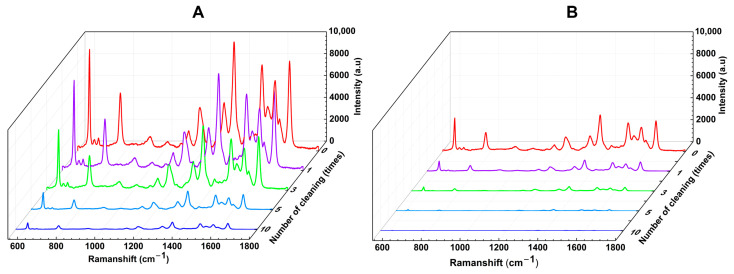
Reusability of the SERS substrates. Raman spectra of R6G after washing the substrate 0, 1, 3, 5, and 10 times for cold plasma and laser treatment (**A**) vs. only cold plasma pre-treatment (**B**).

**Figure 6 nanomaterials-14-01689-f006:**
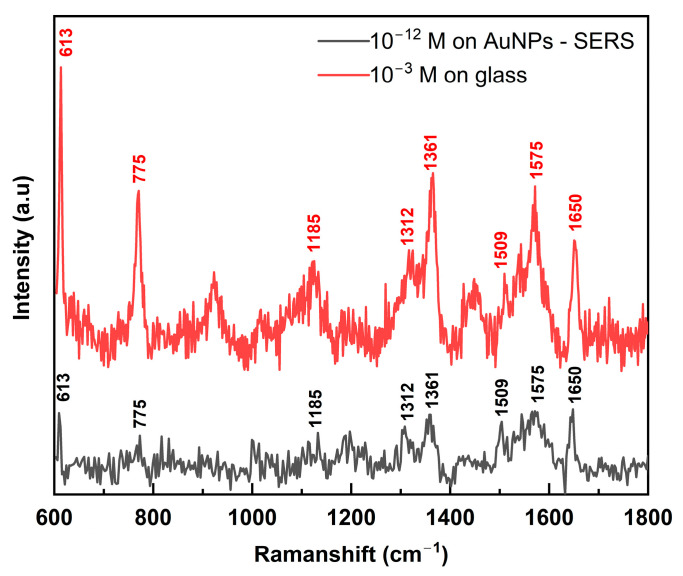
SERS spectra of R6G at 10^−12^ M concentration measured on our fabricated SERS substrate (black) and 10^−3^ M concentration measured on glass substrates.

**Figure 7 nanomaterials-14-01689-f007:**
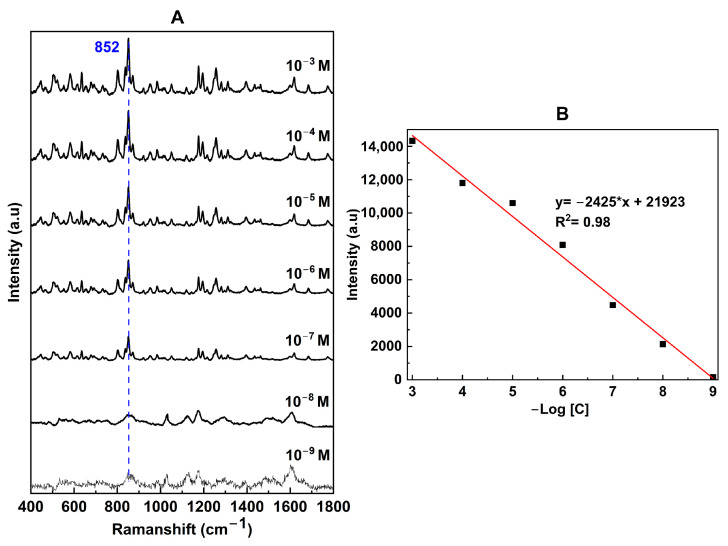
SERS spectra of amoxicillin at different concentrations (**A**) and linear fitting between its concentrations and the peak intensities at 852 cm^−1^ (**B**) over a large concentration range of 10^−3^–10^−9^ M.

**Table 1 nanomaterials-14-01689-t001:** Comparison of fabrication method and enhancement factor of different metal nanoparticle-based SERS substrates.

Substrate Materials	Fabrication Method	Enhancement Factor (EF) Probe Molecule	Ref.
AuNPs/glass (Our work)	Cold plasma treatment + laser deposition	$3.2\times 10^{8}$ (R6G)	
confeito-like Au NPs (100 nm)/Si	Dropping of AuNPs mixture on silicon substrate	10^5^ (R6G)	[28]
AuNPs/silicon nanocrystal/Polystyrene microspheres	Reducing HAuCl_4_ with silicon nanocrystal containing polymer microspheres	5.4 × 10^7^ (4-MPy)	[29]
Nano Au films/Si	Electrochemically roughened nano-Au film	2.45 × 10^8^ (R6G)	[38]
wool roll-like Ag nanoflowers/glass	Mixed ethanol–water reaction	2.7 × 10^6^–5.4 × 10^9^ (R6G)	[30]
AuNPs absorbed on Au nanoplates/Si	Reducing HAuCl_4_ with ascorbic acid and cetyltrimethylammonium bromide (CTAB)	1.7 × 10^7^ (MPH)	[39]

## Data Availability

Data are contained within the article.

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
