# Peer review of "A Novel Method for Rapid and High-Performance SERS Substrate Fabrication by Combination of Cold Plasma and Laser Treatment"

_nanomaterials, 2024, doi:10.3390/nano14211689_

Round 1
Reviewer 1 Report
Comments and Suggestions for Authors
In the present work, authors reported a novel method for the synthesis of high-performance SERS substrate, and then investigated its structural, morphological, and SERS properties. Results indicated that the fabricated SERS substrates exhibit a significant enhancement factor of approximately 3x10⁸ with R6G, allowing detection down to a concentration of 10⁻¹² M. Overall, this work is interesting, and a series of results have also been discussed. However, some issues should be addressed.
1, The abstract can be polished and improved. The novelty problem statement described by the authors should be emphasized to attract general readers by providing more insights on the experimental observations. Also, the authors should elaborate the general applicability of the current work. It was said that “ its application in detecting amoxicillin”.
2, It is well known that there have been numerous studies on Au nanoparticle-based SERS substrates, including their preparation methods and SERS performance. In this manuscript, it would be beneficial to thoroughly explore the advancements and innovations of the Au nanoparticle-based SERS substrate in the introduction section.
3, Additionally, it is important to compare the SERS effects of Au nanoparticle-based substrates in recent years, along with a comparison to commercially available products. Consider utilizing a table format in the discussion section to provide a concise description of these comparisons.
4, Some key and important research results in absorption field should be mentioned and cited so that we can provide a solid background and progress to the readers, such as Nano Research, 2023, 16, 5056; Nano-Micro Letters, 2016, 8, 328.
5, The quality and reliability of SERS substrates are determined by the interaction forces between the substrate and nanoparticles. In this study, cold plasma treatment was employed to enhance the connectivity between the substrate and particles. However, it should be noted that plasma treatment only increases short-term adhesion, and prolonged use or storage may lead to a decrease in adhesion, resulting in a weakened SERS effect. How can we ensure the adhesion of the glass substrate?
6, In SERS substrate materials, factors such as nanoparticle size, inter-particle spacing, morphology, structure, and aggregation degree determine the quality of the SERS effect. How do authors control these structural characteristics of Au nanoparticles in our study? Please provide further details.
7, Based on the experimental results, the detection sensitivity appears to be satisfactory. To reduce costs for commercial applications, future experiments could explore the use of Ag or other more economical metallic nanoparticles as substitutes for Au nanoparticles.
Reviewer 2 Report
Comments and Suggestions for Authors
In this article, the authors report a simple and effective method for rapidly preparing high-performance SERS substrates by combining cold plasma and laser processing. The relevant analysis results indicate that cold plasma pretreatment reduces the roughness of the substrate and increases its surface energy, while laser treatment helps to better uniformly deposit AuNPs to form SERS substrates. This work is very cutting-edge, but there are the following tasks that need to be addressed.
1. What is the physical stability of the SERS substrate? Due to the fact that gold nanoparticles are dropped onto the substrate and have very poor adhesion, the author needs to consider relevant issues.
2. Due to the close correlation between SERS performance and the surface plasmon effect of the substrate. Suggest the author to supplement the UV visible spectral analysis of relevant substrates.
3. All numbers and units in the entire text need to be separated by one space, including in the main text, figures, and tables.
4. There are numerous reports on SERS substrates. What are the advantages of this job compared to existing jobs? Suggest the author to add a table for comparison.
5. The SERS characteristics cannot be separated from the surface plasmon effect. It is suggested that the author provide relevant introductions in the introduction section, and some works need to be mentioned by the author, such as: https://doi.org/10.1016/j.optcom.2024.130816; doi: 10.29026/oea.2023.220072; doi: 10.29026/oes.2022.210008.
6. Further improvement is needed in the English expression related to this job.
Comments on the Quality of English LanguageMinor editing of English language required.
Round 2
Reviewer 2 Report
Comments and Suggestions for Authors
Accept in present form.